# A Blockchain Solution for the Internet of Vehicles with Better Filtering and Adaptive Capabilities

**DOI:** 10.3390/s25041030

**Published:** 2025-02-09

**Authors:** Xueli Shen, Runyu Ma

**Affiliations:** School of Artificial Intelligence, Liaoning University of Engineering and Technology, No. 188 Longwan Street, Sijiatun Street, Xingcheng 125100, China; shenxueli@lntu.edu.cn

**Keywords:** internet of vehicles, blockchain, consensus algorithm, trust management model, gradual acceleration, environmental adaptation, dynamic consensus group

## Abstract

The traditional consensus algorithm based on the Internet of Vehicles (IoV) system has the disadvantages of high latency, low reliability, and weak fault tolerance, and it cannot make real-time adjustments according to the actual environment, making the system vulnerable to malicious control, inefficiency, and poor environmental adaptability. To solve this problem, we propose a gradually accelerating environment adaptive consensus algorithm, AE-PBFT, that can be applied to IoV. It includes a trust management model that achieves gradual acceleration by recording the historical continuous behavior of nodes, thereby improving the efficiency of screening nodes with different intentions, accelerating the consensus process, and reducing latency. At the same time, we introduce a dynamic consensus group division mechanism based on environmental adaptive changes, which can adaptively adjust the number of nodes participating in the consensus process according to the needs of the operating environment, to deal with extreme situations, thereby improving the reliability and fault tolerance of the system. Experiments confirm that the performance of our proposed solution is superior to current solutions in terms of consensus latency and fault tolerance and is more suitable for the operating environment of IoV.

## 1. Introduction

The development of automobiles has gone through several stages, from steam-powered vehicles to modern electric and self-driving cars; automobile intelligence has become an important innovation direction [1,2,3], attracting widespread attention. Smart cars employ advanced information technology, artificial intelligence, and sensor technology, providing a safer, more convenient, and comfortable travel experience. The earliest automobile intelligence concentrated on the car, such as through intelligent driving assistance, in-car entertainment systems, and voice control to improve safety, comfort, and convenience [4,5]. These technologies perform data processing inside the vehicle. With new communication technologies, such as fifth-generation (5G) [6] and the forthcoming sixth-generation (6G) mobile communication [7], vehicles have begun to connect with external entities, bringing unprecedented opportunities and challenges to the automotive industry.

Vehicular Networking [8] is an important component of the intelligent transportation system. It provides a trusted platform connecting vehicles, road infrastructure, and cloud services. Vehicles in the network can interact and integrate valuable information in real time, improving the vehicle’s capabilities in control execution [9], decision-making [10], and environmental perception [11]. However, due to the high speed of vehicles and the dynamic topology of the system caused by sudden interactions, collaboration between vehicles faces security and privacy challenges.

Identity authentication is a critical mechanism for enhancing the robustness of IoV systems and ensuring stable operation [12]. During regular operation, IoV systems may be subject to malicious attacks, such as compromised vehicles intentionally transmitting false information about road conditions to other vehicles, or tampered data at compromised base stations, which can severely compromise the security and reliability of the system. Traditional password-based identity authentication methods, primarily relying on encryption mechanisms like public key infrastructure (PKI), suffer from poor scalability and adaptability. Compared with traditional password authentication, trust management [13] is more flexible, secure, and efficient, and is more suitable for the complex, changeable scenarios of the vehicular network. It can calculate the credibility of messages, and allocate, calculate, and update the reputation values of vehicles, which are generally stored in a central cloud server or roadside unit (RSU). However, the servers storing data are vulnerable to attack [14,15], which may result in the inability to provide reliable and consistent trust services for the entire vehicle network, and can affect the safety of property and passengers.

Blockchain technology is considered a good solution for the above problems. This distributed ledger technology was proposed by Satoshi Nakamoto in 2008 and first implemented with Bitcoin in 2009 [16]. It is based on cryptography and distributed computing, enabling distributed nodes to trade with each other and maintain consistent and tamper-proof ledgers without the supervision of a centralized institution. It links transaction records in blocks in chronological order and uses cryptographic algorithms to ensure they are tamper-proof and secure. Due to its privacy and security characteristics, blockchain has been widely studied and has been applied in non-financial scenarios, such as decentralized storage [17], content delivery [18], and key management [19]. Leveraging the decentralized, tamper-proof, and transparent characteristics of blockchain technology and integrating it with IoV can establish a decentralized and authorized system using the Byzantine consensus algorithm. This integration addresses critical challenges related to data sharing and security.

Although blockchain technology has demonstrated significant potential in enhancing security, privacy, and trust in vehicular networks, it still faces inherent limitations. Issues such as low transaction throughput and high latency, stemming from traditional consensus mechanisms, exacerbate performance constraints in resource-constrained vehicular environments. Additionally, the continuous growth of blockchain ledgers, combined with the limited bandwidth of in-vehicle communication networks, can result in network congestion, reduced throughput, and increased storage costs. Furthermore, most blockchain systems are designed for relatively static networks and struggle to adapt to the rapidly changing topologies of vehicular networks. These challenges collectively hinder the performance, scalability, and adaptability of blockchain technology in such dynamic environments. Several existing studies have proposed various blockchain-based security mechanisms for IoV, introducing lightweight and reputation-based schemes, as well as lightweight consensus protocols built on distributed blockchains. These approaches utilize reputation-proof mechanisms to enhance trust in data sharing among vehicles and partially mitigate these limitations. In practical applications, it is feasible to implement a collaborative and consistent database within RSUs. This setup not only ensures effective trust value management within IoV but also minimizes the risks posed by external attacks on the overall system.

The AE-PBFT algorithm proposed in this paper is specifically designed for the IoV environment and effectively manages trust-related tasks. It enables vehicles to quickly and in real-time update their trust values, thereby enhancing the role differentiation among nodes and improving both consensus efficiency and overall system performance. This paper makes the following contributions:We propose AE-PBFT, an algorithm suitable for IoV, which includes a new credit model and a dynamic consensus group mechanism. The algorithm outperforms previous algorithms in node screening efficiency and fault tolerance;We propose a model of IoV system based on AE-PBFT, which has higher reliability and security than the previous model;Through the analysis of the results obtained from simulation experiments, it can be seen that the proposed algorithm outperforms previous consensus algorithms in terms of consensus delay, throughput, communication overhead, and fault tolerance, and is more suitable for the operating environment of the IoV system.

The rest of this paper is organized as follows: Section 2 introduces work related to IoV authentication technology, trust management models, and blockchain consensus algorithms. Section 3 introduces the Practical Byzantine Fault Tolerance consensus algorithm(PBFT). Section 4 introduces the improved AE-PBFT algorithm suitable for IoV. Section 5 introduces the IoV model based on the AE-PBFT algorithm. Section 6 relates our theoretical analysis. Section 7 analyzes our experimental results. Section 8 explores the potential applications of the proposed algorithm in various other fields. Section 9 summarizes this paper.

## 2. Related Work

We review relevant literature on authentication technologies and trust management models for IoV, and consensus algorithms combining these with blockchain.

### 2.1. Traditional IoV Authentication Technology

Traditional IoV solutions mainly protect data security through encryption. Bayat et al. [20] used elliptic curves and bilinear pairings to construct signatures, to hide the identity of a message sender, but they could not meet the low-complexity requirements of IoV. In [21], Xi et al. proposed an efficient anonymous authentication method based on zero-knowledge proof (ZKP) and elliptic curve cryptography (ECC), where a third trusted institution can effectively track users through their verification keys. In [22], Liu et al. designed an accelerated key negotiation scheme for transmitting sensitive messages, which greatly improved the efficiency of RSU signature verification. In [23], Maurya et al. proposed a secure and efficient conditional privacy anonymous batch authentication scheme (EABAS-CP) based on elliptic curve cryptography, which was suitable for IoV environments. This scheme supports batch signature verification, and the random oracle model (RoM) can use the elliptic curve discrete logarithm problem (ECDLP) to provide unforgeability against adaptive message attacks. Although the aforementioned solutions are highly secure and well-established, the difficulty of management significantly increases as the system scales. Additionally, the drawbacks of high computational resource consumption and prolonged latency render it unsuitable for the dynamic network topology characteristic of the IoV application environment.

### 2.2. Progress of IoV Trust Management Model

Compared with traditional identity authentication methods, the low latency and low computing power consumption of trust management solutions are more adaptable to complex real-world environments. In [24], Yang et al. proposed a trust model using a Dirichlet distribution, reputation regression, and revocation penalty to objectively and accurately reflect the trust status of vehicles, improving the accuracy of malicious vehicle detection and the anti-attack capability of networked vehicles. In [25], Yang et al. used a Bayesian inference model to verify the messages received from adjacent vehicles. An RSU calculated the trust value offset of related vehicles based on the ratings uploaded by the vehicle and packaged these data into blocks. Experimental results showed that the system is effective and feasible in collecting, calculating, and storing the trust values of vehicle networks. In [26], Wang et al. proposed a trust management mechanism, MESMERIC, based on machine learning, which considers direct trust, indirect trust, and context, and separates trustworthy and untrustworthy vehicles through an optimal decision boundary, achieving performance superior to that of other trust management models. In [27], Du et al. designed an incentive mechanism based on the trust value management method of alliance blockchain to encourage vehicles to actively participate in blockchain maintenance. In [28], Jin et al. designed a powerful trust management system by integrating blockchain with the IoV. Rajkumar, V. et al. proposed a novel approach [29] that synergistically integrates an adaptive particle convergence optimization algorithm with a blockchain-based architecture to enhance data trust and alleviate congestion, thereby paving the way for efficient, secure, and reliable vehicular communication networks. A dual-layer blockchain trust management (DLBTM) mechanism was proposed by Ruan et al. [30], which uses logistic regression to accurately calculate the trust value of a vehicle and predicts the probability that the vehicle will provide satisfactory service to other nodes in the next stage. Liu et al. [31] proposed an improved multi-source multi-weight subjective logic algorithm, which integrates the direct and indirect opinion feedback of nodes through a subjective logic trust model and considers factors, such as event validity, familiarity, timeliness, and trajectory similarity, to ensure the security of data storage and sharing. Tripathi et al. [32] proposed a reliability and recommendation (ReTrust)-based scheme, which uses a combined trust model to detect rogue nodes in an IoV network and evaluate the credibility of vehicles. In [33], Wei et al. proposed an architecture to manage the reputation of terminal devices in an Internet of Things (IoT) system and deploy devices based on their locations for data management. The credit management model has significantly advanced research in IoV due to its low resource consumption and strong dynamic adaptability. However, existing solutions primarily emphasize dynamic adaptability and incentive mechanisms, while failing to adequately address issues related to data security storage and overall system security.

### 2.3. Advancements in Consensus Algorithms for Integrated Systems

The combination of blockchain and IoV solves the problem of data storage security, while the consensus algorithm is the key to ensuring the efficient operation of blockchain systems. In [34], Amritesh Kumar et al. proposed a fast and intelligent consensus mechanism, R-PBFT, which improved the PBFT consensus process using a reputation mechanism calculated by logistic regression to achieve a higher performance level. In [35], Xu et al. designed a consensus algorithm ABC-GSPBFT with a group scoring mechanism and an artificial bee colony optimization consensus process to build a consortium chain for sharing flight operation data. In [36], Si et al. proposed an improved practical Byzantine fault-tolerant consensus algorithm, DK-PBFT, introducing a dynamic unique node list to adapt to the dynamic characteristics of IoV. In [37], Tu et al. proposed a vehicle-based secure blockchain consensus algorithm VBSBC to overcome the limitations and shortcomings of the most advanced methods. In [38], Tandon et al. proposed a decentralized architecture based on dual blockchains, which use different consensus algorithms to add another layer of security. In [39], Zhang et al. proposed a consensus algorithm NR-PBFT based on node grouping and reputation evaluation and designed an improved node grouping scheme and reputation evaluation model, which reduced the number of nodes participating in the consensus process while increasing the enthusiasm of nodes to participate in the consensus, greatly reducing communication overhead and transaction delays. In [40], Wang, N et al. proposed a hybrid proof-of-stake blockchain consensus algorithm, and built a secure and efficient distributed IoT data-sharing scheme based on it, enabling each entity to participate equally in the blockchain consensus without consuming high computing power, achieving a higher degree of decentralization. The optimized blockchain consensus algorithm effectively reduces communication delays and costs. When combined with the credit model in IoV, it enhances dynamic adaptability to meet the demands of real-world environments. The system framework based on the AE-PBFT consensus algorithm proposed in this paper is designed in accordance with this principle.

## 3. About PBFT

As the foundation and core of blockchain technology, the consensus algorithm determines how the cluster nodes reach an agreement on the execution order and content of transactions, and ensures the consistency of the node ledger data. Common consensus algorithms include two categories: (1) the proof algorithm, such as proof of work (PoW), and proof of stake (PoS) and its variant, the delegated proof of stake (DPoS), in which miner nodes must win a certain proof through competition to obtain the right to record accounts; (2) the non-proof algorithm, such as PBFT, Paxos, and Raft, in which the system must reach consensus such as through voting or incentives. Because the PBFT consensus algorithm has the characteristics of fault tolerance, high efficiency, and security, it is more suitable for the application environment of IoV. We focus here on the principle process of the PBFT consensus algorithm, analyze its shortcomings, and propose our solutions.

### 3.1. Algorithm Principle and Process

As a state machine replication algorithm, PBFT ensures that consensus can be accurately reached in distributed systems, even in the presence of a certain number of faulty or malicious nodes, thereby achieving consistency in message replication and verification. The feasibility of PBFT is based on the following assumptions: if the number of Byzantine nodes in the system is *f* and the total number of nodes is greater than 3f, i.e., N≥3f+1, then reliable message transmission can be carried out between nodes. Its key idea is to reach consensus through the collaboration of multiple replicas. At each stage, a sufficient number of messages must be collected to ensure security and correctness. The process of the algorithm, as shown in Figure 1, has the following phases:

(1) Request. The client sends a transaction request $\langle Request,o,t,c\rangle$ to the cluster’s Primary node, where *o* represents the specific operation or command that the client wants to perform, *t* represents the timestamp and *c* represents the client’s unique identifier. If the request is sent to a Replica node, it will be forwarded to the Primary node;

(2) Pre-Prepare. After receiving the transaction request, the Primary node performs verification, packaging, and sorting operations on the transaction. After confirming that it is correct, it will broadcast $\langle Pre-Prepare,v,n,d\rangle$ to the Replica node, where *v* represents the current view number, *n* represents the sequence number and *d* represents the content summary;

(3) Prepare. The Replica node checks the legitimacy of the Pre-Prepare message sent by the Primary node. If the check passes, it broadcasts the Prepare message $\langle Prepare,v,n,d,n\rangle$ to other consensus nodes for cross-verification. If the check fails, the message will not be sent. Note that if most nodes in the system fail the check at this time, and therefore, do not send the Prepare message, resulting in a transaction timeout, the transaction will trigger a view-switching operation;

(4) Commit. The Replica node checks the consistency of the received Prepare message with the Pre-Prepare message content in the cache. If a sufficient number 2f+1 of Prepare messages that meet the requirements are received, the Commit message $\langle Commit,v,n,n\rangle$ will be broadcast to indicate that this node can execute the transaction;

(5) Reply. The Replica slave node performs a consistency check on the received Commit message and the Prepare message content in the cache. If the check passes, the execution result $\langle Result,v,t,c\rangle \left\{n\right\}$ will be written to the local ledger and replied to the client. If the client can receive f+1 enough consistent replies within the timeout period, then the transaction has been successfully completed.

### 3.2. Disadvantages of Original Algorithm

(1) All nodes in the system must participate in the consensus process. Although it is fair, it will occupy computing resources and reduce consensus efficiency. Therefore, a better consensus node selection strategy is needed;

(2) Selecting the master node in the sequence has a high security risk, and attackers will exploit this feature;

(3) When the master node fails or behaves maliciously, a view switch is required to select a new master node, which requires broadcasting and collecting additional messages, which takes time to complete, causing system pauses and performance degradation.

## 4. Gradually Accelerating Environment Adaptive Consensus Algorithm

In view of the high performance cost of traditional consensus algorithms, such as PBFT, this paper proposes an improved AE-PBFT algorithm based on a gradually accelerated credit management mechanism and an environment-adaptive consensus group dynamic division mechanism. As shown in Figure 2, the AE-PBFT algorithm has the following steps:

(1) All nodes N are initialized in the system. The client sends a transaction request $\langle Request,o,t,c\rangle$ to the cluster master node, which contains operations and parameters;

(2) The master node broadcasts the received request $\langle Pre-Prepare,v,n,d\rangle$ to all backup nodes, which verify the legitimacy of the request and broadcast $\langle Prepare,v,n,d,n\rangle$ to other backup nodes after confirmation;

(3) After receiving a request broadcast by other backup nodes, a backup node verifies the legitimacy of the request. If the verification passes, the node enters the response phase;

(4) All nodes entering the response phase verify the message and send the result to the master node;

(5) When the master node receives more than 2f+1 identical confirmation messages, it believes that the system has reached a consensus, sends the confirmation result $\langle Result,v,t,c\rangle$ to all nodes, and synchronously updates the node trust score. At this point, the consensus process ends.

In view of the slow screening speed of the traditional fixed plus and minus trust model, this paper proposes a new trust model and consensus group division mechanism, which can quickly and effectively distinguish malicious nodes and divide consensus groups. There are two core mechanisms in AE-PBFT: the node trust management model and the consensus group division mechanism.

### 4.1. Node Trust Management Model Based on Gradual Acceleration Mechanism

#### 4.1.1. About the Learning Rate

In deep learning, the concept of the learning rate comes from model training in the gradient descent method, which is the basic optimization algorithm for training deep neural networks. It minimizes the loss function by updating parameters to achieve model learning and optimization.

The gradient descent iteration formula is as follows:(1)$$\Theta^{n+1}=\Theta^{n}-\alpha_{learn}*\frac{\partial J\left(\theta \right)}{\partial \theta },$$
where $\Theta^{n+1}$ and $\Theta^{n}$ denote the model parameters to be optimized at the (n+1)th and *n*th training batches, respectively; $\alpha_{learn}$ represents the learning rate (or step size), and $J\left(\theta \right)$ is the loss function. The choice of learning rate $\alpha_{learn}$ determines the convergence speed of model training. When $\alpha_{learn}$ is too low, the loss function value will decrease slowly each iteration, which will extend the training time and bring additional performance overhead. Only when the learning rate increases to a suitable value will model training be faster and perform better. The decline rate of the loss function under different learning rates is shown in Figure 3. Many variant optimization algorithms have been proposed, such as RMSprop (historical gradient mean), Adam (adaptive moment estimation), and Mini-batch Gradient Descent.

Inspired by the concept of learning rate, we have improved the traditional trust value model of IoV. To address the problems in quickly identifying malicious vehicles and screening consensus vehicles, we propose a model that uses a gradually accelerating method to expose trusted and malicious vehicles. We introduce its basic principles and implementation methods.

#### 4.1.2. Trust Management Model Based on Gradual Acceleration Mechanism

To intuitively describe the algorithm model we proposed, the parameter definition was performed in Table 1.

As an efficient reward mechanism, the trust management model has significant advantages, such as strong scalability, high node participation, and high quality of information interaction. However, the traditional trust model only focuses on the improvement of the algorithm on the existing mechanism and does not take into account Credit score update efficiency and node screening speed. The proposed trust model based on the gradual acceleration mechanism largely solves problems of slow response to malicious behaviors and a long delay in determining the role of normal vehicles. Figure 4 shows a flowchart of the method, which has the following process:

(1) System Initialization: Initialize the system nodes N, initial credit Scores [$N_{Node\_id}$], and the consensus group ratio $P_{0}$. At this point, the order of all nodes is randomly determined, and the system parameters are initialized. These include the reward weight coefficient α, penalty weight coefficient β, trust coefficient $W_{t}$ based on historical behavior $W_{h}$, and the continuous historical behaviors $\xi_{1}$ and $\xi_{2}$.

(2) Master Node Selection: Based on the master node selection rule $n_{0}=v\;mod\;len\left(N_{CO}\right)$, the master node for the current stage is selected. The client then sends a request to the master node $n_{0}$ to initiate the consensus process, as outlined in Figure 2.

(3) Reply Phase: During this phase, the master node $n_{0}$ provides the confirmation result to all nodes, including both consensus nodes $N_{CO}$ and candidate nodes $N_{CA}$. Simultaneously, the credit Scores of all participating nodes are updated.

(4) Credit Score Update: The RSU updates the parameters in the two counters $W_{h}$ and $W_{t}$ based on the correctness of the received results. If a node’s judgment aligns with the final result, the corresponding $\xi_{1}$ in $W_{h}$ for that node increases by 1; otherwise, $\xi_{2}$ decreases by 1. When $\xi_{1}$ or $\xi_{2}$ reaches a threshold of ≥3 or ≤−3, the system considers the node to be in a consistent behavioral state and adjusts the weight coefficient in the $W_{t}$ counter. This adjustment influences the node’s credit score settlement in the current round, thereby accelerating the identification and classification of nodes with different roles.

(5) Consensus Group Adjustment: The RSU calculates the current overall average credit score of all nodes to assess the impact of external environmental factors. Since the consensus group directly participates in the consensus process, its weight in the calculation is higher. The new consensus group ratio *P* for the next round is determined using Equation (Equation 3), which helps to eliminate nodes that disrupt normal operations. All updated information will be recorded on the blockchain as permanent data storage.

(6) Next Consensus Round: The credit scores and the composition of the consensus group are updated, and the system proceeds to the next round of the consensus cycle.

As shown in Algorithm 1, to track the continuous historical behavior of nodes, we have established a counter dictionary, $W_{h}$, to record each node’s behavior. For each node ID $N_{Node\_id}$, it stores two count values: $\xi_{1}$ and $\xi_{2}$. $\xi_{1}$ tracks the cumulative score increases for the node, starting at 0 and incrementing by 1 with each increase. Conversely, $\xi_{2}$ tracks the cumulative score decreases, starting at 0 and decrementing by 1 with each decrease. When either $\xi_{1}$ reaches 3 or $\xi_{2}$ reaches −3, it indicates that the node has exhibited continuous increasing or decreasing behavior three times. At this point, the node is considered to have demonstrated a consistent pattern of correct or incorrect behavior. We then gradually adjust the node’s reward weight coefficient α and penalty weight coefficient β in the trust coefficient counter $W_{t}$, which corresponds to the node. Both coefficients initially start at 1, and as the node continues to exhibit consistent behavior, these values increase, up to a maximum of 4. The opposing coefficient (reward or penalty) will remain unchanged. The specific rules are outlined in Table 2 and Table 3. This means that in each round of credit score settlement, a node’s credit score can change up to four times faster than the initial rate, depending on its behavior. Nodes with a history of positive behavior will see their credit scores increase more rapidly, while those with a history of malicious behavior will experience faster declines.

Unlike the fixed scoring model of traditional credit systems, this dynamic mechanism allows for a quicker divergence in scores between nodes of different behavioral patterns. As a result, the system becomes more efficient at role selection, which indirectly enhances the robustness of the IoV system.
**Algorithm 1** Trust Management Model Based on Gradual Acceleration Mechanism**Require:** Reward weight coefficient *α*, Penalty weight coefficient *β*, System Nodes N,   historical_behavior $W_{h}$ = {}, trust_coefficient $W_{t}$ = {}.
 1:System initialization 2:α = 1, β = 1 3:$W_{h}$[N] = [0, 0], $W_{t}$[N] = [1, 1] 4:Note that the parameter $W_{h}$[$N_{Node\_id}$][0] is here equal to $\xi_{1}$ and that $W_{h}$[$N_{Node\_id}$][1] is $\xi_{2}$. 5:Complete the consensus process to the reply stage 6:Start update 7:**if** result == accepted_response **then** 8:    $W_{h}$[$N_{Node\_id}$][0] += 1 9:    **if** $W_{h}$[$N_{Node\_id}$][1] < 0 **then**10:        $W_{h}$[$N_{Node\_id}$][1] = 011:        $W_{t}$[$N_{Node\_id}$][1] = 112:    If there is a continuous bonus behavior, the reward weight coefficient $W_{t}$[$N_{Node\_id}$][0] will continue to increase13:    $\tilde{a}\leftarrow W_{t}\left[N_{Node\_id}\right][$0]14:    **if** $\mathrm{Scores}[N_{Node\_id}]<90$ **then**15:        $\tilde{a}\leftarrow 4$16:    Scores[$N_{Node\_id}$] += $\tilde{a}$ *117:**else**18:    $W_{h}$[$N_{Node\_id}$][1] -= 119:    **if** $W_{h}$[$N_{Node\_id}$][0] > 0 **then**20:        $W_{h}$[$N_{Node\_id}$][0] = 021:        $W_{t}$[$N_{Node\_id}$][0] = 1;22:    If there is a continuous deduction behavior, the penalty weight coefficient $W_{t}$[$N_{Node\_id}$][1] will continue to increase23:    $\tilde{b}\leftarrow W_{t}\left[N_{Node\_id}\right][$1]24:    Scores[$N_{Node\_id}$] -= $\tilde{b}$ *125:Update completed

It is important to note that when a node is in an accelerated credit score accumulation phase due to continuous correct historical behavior and it suddenly performs an incorrect action, how should this accelerated state be adjusted? Our proposed solution is to interrupt the process and revert the node to its initial state. That is, once a behavior contrary to the current pattern is observed, all accumulated state benefits for the node are canceled, and the accumulation process must restart from the beginning. This is because the continuous behavior has been disrupted by the anomalous action, necessitating a fresh accumulation process. We implement a simulated semaphore mechanism to achieve mutually exclusive recording of node historical behavior, which will be discussed in detail in the next section.

#### 4.1.3. Recording Historical Continuous Behavior by Simulating the Mutual Exclusion Semaphore Mechanism

To avoid multiple threads simultaneously occupying the same system resources and causing interlocking in the operating system, a recording semaphore mechanism is proposed, as shown in Figure 5. By setting a mutual exclusion semaphore and performing PV operations before and after a thread enters the critical section, the plannable use of limited system resources by the thread is achieved.

We designed a simulated mutual exclusion semaphore mechanism, as illustrated in Figure 6, to track the historical continuous changes in node credit scores. When one of the count value parameters in a node’s history ($W_{h}$) accumulates to a non-zero value and another count value suddenly changes, it indicates that the node’s historical behavior has been interrupted. At this point, following the mutual exclusion mechanism, we reset the accumulated value to zero, and all parameters begin counting anew. For instance, when a node is verified as correct and its credit score needs to be increased, $\xi_{1}$ is typically incremented by 1, $\xi_{2}$ is immediately reset to 0, and β is initialized to 1. This indicates that the previous continuous deduction behavior has been interrupted, prompting all parameters to start counting again. The advantage of this approach is that it ensures data consistency and mitigates potential errors that may arise from complex concurrent parameter updates.

#### 4.1.4. Low Score Compensation Mechanism

Due to continuous point deductions caused by factors, such as traffic congestion or master node failures, non-malicious nodes may remain abnormally offline and unable to participate in consensus. To encourage the engagement of normal nodes in the consensus process, we have established an early warning threshold $H_{Threshold}$. For nodes with credit scores falling below this threshold, their reward weight coefficient α will temporarily increase to the maximum value until they exit the low score early warning state, while the penalty weight coefficient β remains unchanged. This approach ensures that malicious nodes, even if they occasionally exhibit honest behavior, will find it difficult to reintegrate into the consensus process. The specific rules are detailed in Table 4.

In this section, we establish a scenario to determine the value of the warning threshold. An honest node enters an unresponsive state for a period following the system startup due to an accident. During this period, our proposed solution classifies it as a malicious node exhibiting continuous malicious behavior. Consequently, its $\xi_{2}$ value continues to decrease while β gradually increases. By the sixth round of the consensus process, $\xi_{2}$ has reached −5, and β has risen to its maximum value of 4. We assume this represents the upper limit of tolerance for an honest node experiencing a cold start; that is, below this score, the node will not be deemed as exhibiting malicious behavior unintentionally. The calculated value is 88, leading us to set $H_{Threshold}$ at 90, thereby allowing for a degree of fault tolerance (indicating a correct response but with very weak performance).

Considering the complex conditions vehicles encounter during actual operation, it is not uncommon for credit scores to drop unexpectedly due to factors, such as traffic congestion, adverse weather, or system maintenance. The occurrence of these situations does not inherently indicate malicious behavior. To encourage vehicle nodes to actively participate in the consensus process while maintaining appropriate fault tolerance, we opted not to establish a threshold for permanent exclusion.

### 4.2. Dynamic Consensus Group Division Mechanism Based on Environmental Adaptive Changes

All nodes in PBFT participate in the consensus process, at the cost of increased communication overhead and time complexity. The consensus group mechanism only selects a portion of the reliable nodes to participate in the consensus, which improves system operation efficiency. Most traditional consensus groups adopt a fixed ratio and cannot dynamically adapt to the actual operating environment. This may cause instability, which may be exploited by malicious nodes. We propose a dynamic consensus group division mechanism based on environmental adaptive changes in Algorithm 2. It has been proven that nodes can be effectively screened in a complex operating environment to realize higher performance.

The new consensus group division mechanism is improved on the basis of the original fixed ratio. When updating the node credit scores at the end of each round of consensus, those of the consensus $N_{CO}$ and candidate groups $N_{CA}$ are counted separately, and weighted aggregation is used to form the overall mean parameter. These data reflect the quality of the current actual operating environment. This parameter is brought into a monotonically increasing inverse trigonometric function to obtain the consensus group ratio *P*, which adapts to environmental changes. We choose to update the ratio data every 5–10 rounds and apply this change to the system to enhance its anti-interference ability and fault tolerance.

The weighted aggregate mean of the credit scores of all nodes in the system is as follows:(2)$$\bar{x}=\frac{x_{1}*a_{1}+x_{2}*a_{2}}{n_{1}+n_{2}},$$
where $x_{1}$ and $x_{2}$ are the respective sums of scores of consensus and candidate group nodes, with respective weighted statistical coefficients of $a_{1}$ and $a_{2}$, which we set to 0.6 and 0.4, respectively. The consensus group carries greater weight than the candidate group because it participates more actively in the consensus process, thereby exerting a more significant influence on the security and stability of the system. $n_{1}$ and $n_{2}$ are the respective numbers of consensus and candidate group nodes.

The proportion of the consensus group to the entire node is as follows:(3)P=1π∗arctanx¯−100∗45+25,
where x¯ is the average value of the entire node. As a monotonically increasing inverse trigonometric function ranging from 0 to 4/5, its change amplitude is most drastic at the initial value of 2/5 and is relatively small near 0 and 4/5. This is consistent with our actual operating environment, i.e., when the system is initialized, the consensus group proportion coefficient caused by the fluctuation of the overall behavior of the vehicle changes the fastest. When the accumulation time is long, the change in this coefficient is slower.
**Algorithm 2** Dynamic consensus group partitioning mechanism based on adaptive environmental changes**Require:** Consensus group proportion coefficient P0, consensus_nodes NCO,   candidate_nodes NCA, weight_coefficient *a*.**Ensure:** Consensus group proportion coefficient *P*.
 1:The consensus process ends and the update process begins. 2:a1 = 0.6, a2 = 0.4 3:n1 = len(NCO), n2 = len(NCA) 4:x1 and x2 are, respectively, the sum of node scores within the consensus group and candidate group. 5:x¯=x1∗a1+x2∗a2n1+n2 6:P=1πarctanx¯−100∗45+25 7:Update completed.

Although the 2/5 consensus group ratio may increase the likelihood of Byzantine nodes participating in the consensus and disrupting the normal operation of the system, the rapid iteration speed provided by the AE-PBFT algorithm enhances the efficiency of node screening. Even if malicious nodes are included in the consensus group initially, they cannot remain for an extended period. Furthermore, in practical applications, the performance requirements of the IoV system are often higher, making it worthwhile to make certain compromises in security for the benefits of reduced latency and lower energy consumption.

## 5. IoV System Model Based on AE-PBFT Algorithm

### 5.1. System Model

As shown in Figure 7, the IoV system applicable to the proposed AE-PBFT algorithm is a multi-layered model, including layers for Certificate Authority (CA) authentication, edge interaction, and data storage.

(1) CA authentication layer. The IoV system using consortium chain technology has the advantage that the identities of all members are verified to be authentic and reliable, which greatly reduces the cost of achieving trust. The CA authentication layer includes the CA center and the cloud server. The CA center records and verifies the identities of each entity, such as issuing identity certificates and issuing, managing, and destroying keys for RSUs and on-board units (OBUs). The cloud server stores and manages the identity data of each entity and verifies the legitimacy and timeliness of audit records. The CA center places entities that engage in malicious behavior on the list of dishonest persons and broadcasts them to the IoV system through 5G NR-V2X communication technology, prohibiting them from participating in subsequent operations;

(2) Edge Interaction Layer. This layer includes an OBU and RSU. Vehicles equipped with various sensors and unit modules act as mobile nodes and need to send messages to the surroundings and receive them in real time. Interaction methods include vehicle-to-vehicle (V2V) and vehicle-to-roadside (V2R) units. V2V establishes a trusted collaboration channel between vehicles, while V2R establishes relevant service request and response channels;

(3) Data Storage Layer. The collected trust values and rating information will be uploaded to the distributed blockchain database to ensure permanent and secure data storage. In contrast, traditional centralized database storage methods are more susceptible to attacks, which can result in data loss or tampering.

### 5.2. Identity Authentication

(1) During the parameter initialization phase, the system generates public–private key pairs in batches using the ECC asymmetric encryption algorithm to facilitate the distribution and management required for subsequent registration steps. A random number $P_{private}$ is selected as the private key on a non-singular elliptic curve *E*, and its corresponding public key $P_{public}$ is calculated accordingly.

(2) When new vehicles are added to the system, they must initiate a registration identity request to the nearby RSU. The RSU collects and encrypts the vehicle’s identity information, then transmits it to the cloud server for processing and backup. Subsequently, the information is submitted to the CA for review and certification. Assuming the vehicle set is *V* = $v_{1}$, $v_{2}$, $v_{3}$ … and the RSU set is *R* = $r_{1}$, $r_{2}$, $r_{3}$ …, the CA center reviews the identity request of each vehicle. For vehicles that pass the review, the CA issues an identity certificate in the form of ($T_{i}$, ${PW}_{i}$, ${DID}_{i}$), where $T_{i}$ is the registration timestamp, ${PW}_{i}$ is the password assigned to the vehicle, and ${DID}_{i}$ is a unique decentralized identifier. Additionally, the CA maintains a verifiable data registry (VDR) *W* = ${DID}_{1}$, ${DID}_{2}$, ${DID}_{3}$ …, which records all registered DIDs. The CA only issues certificates to DIDs that are registered in the VDR. If a vehicle’s DID is not found in the registry, it indicates that the identity request has failed the review process, and the vehicle cannot be added to the IoV system.

(3) The RSU receives the vehicle registration message from the CA center and verifies it. At this point, the RSU acts as a node on the blockchain. It employs the blockchain consensus algorithm to determine whether consensus can be reached. If the RSU decides to trust the vehicle, it will grant the vehicle a digital signature; otherwise, it will refuse to record the vehicle information in the blockchain.

### 5.3. Possible Attacks

The IoV system faces a variety of malicious threats in real-world operating environments. Compared to external attacks, internal threats are more challenging to detect and prevent, as they often originate from within the system and exploit its trust-based architecture. Attackers can undermine network stability and disrupt the trust foundation of IoV through various means, compromising the safety and efficiency of vehicular communication and decision-making. Below, we outline and analyze four common types of attacks in IoV environment:

(1) Direct attacks from malicious vehicles: Malicious vehicles can directly send false or misleading messages to nearby vehicles. These false messages may include incorrect information about traffic conditions, road hazards, or vehicle positions, leading surrounding vehicles to make inappropriate or dangerous decisions. For example, a malicious vehicle could falsely broadcast the presence of a traffic jam or an accident ahead, causing other vehicles to reroute unnecessarily, leading to congestion on alternate paths. Alternatively, it could send incorrect positioning data, resulting in collisions or traffic disruptions. Such attacks exploit the decentralized nature of IoV, where vehicles rely heavily on real-time data from peers to make autonomous driving decisions. Detecting these attacks can be challenging, as the malicious messages may appear legitimate, and their impact can propagate quickly through the network;

(2) Indirect attacks from malicious vehicles: In this type of attack, malicious vehicles do not directly interfere with other vehicles but instead target the reputation and trustworthiness of normal vehicles. They achieve this by sending false evaluations of legitimate vehicles to RSU, which is responsible for maintaining trust records and calculating credibility scores. By distorting the credibility judgments of normal vehicles, attackers can manipulate the system to distrust reliable vehicles, thereby disrupting the normal operation of IoV. For instance, a malicious vehicle may report that a well-behaved vehicle has been sending incorrect data or behaving erratically. This can lead to the legitimate vehicle being excluded from consensus processes or other cooperative tasks within the network. Over time, such manipulation can erode the reliability of IoV’s trust mechanisms and lead to cascading system failures;

(3) Direct attack on RSU: As a critical component of IoV, the RSU serves as a local hub for communication, data storage, and trust management. Direct attacks on an RSU involve an attacker infiltrating the unit, either physically or through network vulnerabilities, to manipulate its stored data. This may include altering historical credibility evaluations, tampering with trust scores, or modifying vehicle-related data. For example, an attacker could modify the stored trust score of a malicious vehicle to make it appear trustworthy, allowing it to participate in consensus or mislead other vehicles. Similarly, the attacker could delete or modify records of legitimate vehicles, undermining their credibility and potentially excluding them from the network. Such attacks can have a widespread impact, as the RSU often serves multiple vehicles within its communication range, and compromised data can propagate through the network;

(4) Indirect attack on RSU: Unlike direct attacks, indirect attacks on the RSU target its internal logic rather than its stored data. In an indirect attack, the attacker modifies the algorithms or parameters used by the RSU to calculate vehicle credibility. For example, the attacker could alter the weighting of specific trust metrics or introduce biased parameters that favor malicious vehicles. This manipulation could render the RSU incapable of accurately evaluating the trustworthiness of vehicles, allowing malicious nodes to evade detection while penalizing legitimate vehicles. Such attacks are particularly dangerous because they undermine the foundational processes that maintain trust in IoV. Without accurate trust calculations, the entire system’s reliability is compromised, leading to widespread disruption in vehicular communication and decision-making.

### 5.4. Security Requirements

The system should meet the following security requirements during operation:

(1) Privacy security. The identity of a vehicle should be protected from third parties to prevent information leakage;

(2) Authentication legitimacy. During network transmission, the receiver should ensure that the complete message can be received, and the legitimacy of the sender’s identity must be guaranteed;

(3) Movement trajectory protection. The movement trajectory of a vehicle must be protected to prevent attackers from inferring the owner’s private information;

(4) Anti-attack. The system must be able to resist the above types of attacks to ensure stability and normal operation.

## 6. Theoretical Analysis

We will perform a theoretical analysis of the proposed solution from three key perspectives: security, communication overhead, and environmental adaptability. The details of each aspect are outlined below.

### 6.1. Security Analysis

AE-PBFT is an enhancement of PBFT that introduces a novel credit model and a consensus group mechanism. It determines the selection of consensus groups and master nodes based on the credit points accumulated through historical behavior. This approach mitigates the risk of the master node and the consensus process being controlled by malicious nodes. The security analysis of the system is as follows:

Assume that the number of vehicle nodes participating in the IoV system is *N*, the number of consensus group nodes is $N_{c}$, and the number of malicious Byzantine nodes is *f*. According to the group division rules of the AE-PBFT algorithm, the relationship between these variables is approximately $N\approx 2N_{c}/5$ and $N_{c}\geq 3f+1$. Rearranging these equations, we derive $N\geq \left(6f+2\right)/5$. Given that there is at least one Byzantine node in the system(f≥1), it follows that N≥8/5. Rounding up, we find that N≥10, meaning that the security of the consensus process can only be assured when the number of participating vehicles in the IoV system is at least 10. This condition is easily met in real-world environments with large traffic volumes, ensuring the security of the algorithm.

Next, we analyze the security of the consensus process. Continuing with the assumption that the number of vehicle nodes in the IoV system is *N* (where N≥10), the list of participating nodes is denoted as $N_{Nodeid}$ = [$n_{0}$, $n_{1}$, $n_{2}$, …, $n_{N-1}$], and the number of malicious Byzantine nodes is *f*. In this analysis, we use simplified stage names to represent the messages received by the nodes.

In the request phase, the master node $n_{0}$ broadcasts a $\langle request\rangle$ message to the replica nodes.

In the pre-prepare phase, each replica node verifies the received message. Based on the known condition N≥3f+1, we can derive the following:(4)$$\sum_{i=1}^{N-1}{\langle pre-prepared\rangle }_{i}\geq 2f.$$

In the prepare phase, each participating node not only sends its own message but also receives messages from other nodes. For each node *i*, the prepared message set can be represented as follows: ${\langle prepared\rangle }_{i}={\langle prepared\rangle }_{0}+{\langle prepared\rangle }_{1}+\ldots +{\langle prepared\rangle }_{i-1}+{\langle prepared\rangle }_{i+1}+\ldots +{\langle prepared\rangle }_{N-1}$. Finally, we can derive the following:(5)$${\langle prepared\rangle }_{i}\geq 2f.$$

In the reply phase, it is established that ${\langle prepared\rangle }_{i}\geq 2f$, indicating that at least 2f nodes can correctly process the message and respond. Consequently, the total number of messages that the master node $n_{0}$ can receive during the reply phase is as follows:(6)$${\langle response\rangle }_{i}\geq 2f+1.$$

At this point, it can be concluded that a security consensus has been achieved.

The AE-PBFT consensus process simplifies the commit broadcast phase compared to PBFT, resulting in a significantly reduced time to reach consensus. In IoV, this allows for the rapid updating of credit scores, enabling the system to effectively distinguish between honest and malicious vehicles. This adaptability is particularly advantageous in dynamic real-time environments where situations change rapidly.

### 6.2. Communication Overhead

The consensus process of the traditional PBFT has three stages. In the pre-preparation stage, the master node sends messages to other backup nodes. At this time, the number of communications in the network is n−1. In the preparation stage, the backup nodes broadcast the preparation message to each other, and the number of communications is $\left(n-1\right)*\left(n-1\right)$. In the submission stage, the backup nodes submit messages to each other for further verification, and the number of communications is $\left(n-1\right)*\left(n-1\right)$. In the reply stage, the backup node replies to the master node, and the number of communications is n−1. Therefore, the communication overhead of PBFT in this process is $2n*\left(n-1\right)$.

The main stages of the GPBFT algorithm [41] are Pre-prepare, Prepare, and GroupSign. In the Pre-prepare stage, the master node broadcasts messages to other nodes n−1 times. In the Prepare stage, each node broadcasts Prepare messages to other nodes $n*\left(n-1\right)$ times. In the GroupSign stage, after the group administrator and other nodes form a group, the node that receives the Prepare message needs to sign a short group signature to respond to the client, so the number of communications is *n*. The total communication overhead is $n^{2}+n-1$.

In CBFT [42], the consensus process is divided into inter-group and intra-group consensus. Assuming *n* nodes, each consensus set has *m* $\left(m\geq 3\right)$ nodes. Each consensus set has two stages, preparation and submission, which require sending messages for communication. The two are n−n/m, respectively. The number of communications within the consensus group is ${\left(\frac{n}{m}\right)}^{2}-n+m$, so the total communication overhead of CBFT is ${\left(\frac{n}{m}\right)}^{2}-\frac{3n}{m}+2n$.

AE-PBFT optimizes the scale of nodes participating in the consensus process, dynamically adjusts the consensus group ratio according to the credit status of the overall nodes, and only requires some trusted nodes to participate in the consensus. Nodes are divided according to the credit values of continuous historical behaviors, eliminating the tedious Commit steps. After multiple verifications, the proportion of the consensus group is stable at 2/5 in the actual operating environment, so the communication overhead is approximately $\left(\frac{2n}{5}-1\right)*\left(\frac{2n}{5}+1\right)$.

Comparative analysis in Table 5 indicates that AE-PBFT, while maintaining security and stability, incurs lower communication overhead compared to the other three algorithms. Although the maximum fluctuation in the consensus group ratio reaches 4/5, it generally remains around 2/5 except in extreme cases. The advantage of a smaller consensus group is that it consumes fewer computing resources from the vehicle processors, enables faster response times, and provides greater flexibility over time. Future optimization directions could focus on the development of smaller, more flexible consensus groups and more adaptive dynamic consensus group adjustment algorithms. These improvements are expected to enhance system performance, particularly in terms of reduced energy consumption and lower latency.

### 6.3. Environmental Adaptability Analysis

Assume that at the beginning of the IoV system operation, there are 10 client nodes with an average credit score of 100. The ratio of consensus group nodes to candidate nodes $P_{0}$ is set to 2/5, meaning that four nodes belong to the consensus group and six nodes belong to the candidate group. When the system encounters various challenges, such as signal interference caused by road congestion, adverse weather conditions, or equipment maintenance, as well as poor network conditions, some nodes may experience performance degradation, leading to a reduction in their credit scores and a decrease in the overall average score.

In this scenario, assuming the current average credit score drops to 99.586, it can be calculated using Equation (Equation 3) that the new ratio *P* is approximately 3/10. This adjustment implies that the number of consensus group nodes decreases to 3, while the number of candidate group nodes increases to 7. Given that the consensus group nodes carry a higher weight in Equation (Equation 2), it suggests that some consensus group nodes are more likely to underperform or fail under challenging conditions. Thus, dynamically reducing the size of the consensus group helps retain as many functioning nodes as possible, allowing the system to operate more reliably and maintain better adaptability to environmental changes.

## 7. Simulation Experiments and Analysis of Results

We conducted simulations of vehicular communication within the IoV environment to evaluate the performance of various consensus algorithms. For simplicity, we assumed that the IoV operating environment is sufficiently open, and the distances between vehicles and related infrastructure are large enough, allowing the vehicles to be modeled as point objects in a relatively static state. In our experimental setup, consortium blockchain nodes were used to represent the vehicles for simulation purposes. The experiments were implemented using Python 3.8 on the PyCharm platform, running on a machine equipped with a 12th-generation Intel Core i5-1240P processor and 16 GB of RAM. We simulated the performance of four consensus algorithms: PBFT, GPBFT, CBFT, and AE-PBFT. GPBFT and CBFT are two benchmark algorithms used for experimental comparison. GPBFT is a Byzantine fault-tolerant consensus algorithm that leverages short group signatures with a two-level administrative structure. A certification authority classifies node identities based on credit values and employs group signature schemes to trace the true identities of malicious nodes, thereby enhancing the system’s security and stability. The CBFT algorithm, on the other hand, divides the consensus process into inter-group and intra-group consensus based on the network nodes’ response speed to management nodes. It further categorizes nodes into different types using a credit model to minimize the likelihood of the master node being malicious. Compared to PBFT, these two algorithms achieve significant improvements in communication overhead and security, making them more robust and suitable for experimental comparison. Successful program execution, indicated by correct output, signified that consensus was achieved. To assess the performance of these algorithms, we varied the number of nodes in the network from 0 to 100 and evaluated key metrics, including consensus latency, throughput, communication overhead, and fault tolerance.

### 7.1. Consensus Latency

First, we will assess the consensus latency of the four algorithms. Consensus latency is defined as the time interval between when a client issues a request and when it receives a reply. As a critical performance indicator for blockchain systems, consensus latency directly impacts user experience, real-time system performance, throughput, and security. Low-latency blockchain systems offer significant advantages in various application scenarios, providing more efficient and reliable services. We fixed the number of requests at 200 and conducted 100 transaction tests across a range of 0 to 100 nodes in increments of 10, calculating the average as the result. This outcome will serve as the basis for subsequent analysis. The consensus delay is calculated as follows:(7)$$T_{D}=t_{s}-t_{r},$$
where $t_{s}$ is the time when the client sends a request, and $t_{r}$ is the time when a reply is received. The experimental results are shown in Figure 8.

As the number of participating clients increases, the consensus delay of all four algorithms correspondingly rises. However, the increase in delay for the AE-PBFT algorithm is the least pronounced. The delay times presented in the data of Table 6 are significantly lower compared to the other schemes. On average, it is reduced by about 98.02% compared with PBFT, GPBFT, and CBFT. This improvement is attributed to the new approach, which incorporates only one broadcast stage in the process, thereby reducing the volume of information exchanged. Additionally, by utilizing a small-scale consensus group that is updated rapidly in real-time, the number of nodes participating in the consensus is minimized, enhancing adaptability in real-world environments.

### 7.2. Throughput

In this section, we will conduct a throughput experiment. Throughput refers to the number of transactions completed per unit of time. Higher throughput indicates better scalability, concurrency, and data processing capabilities. We set the number of requests at 200 and tested 100 transactions across a range of 0 to 100 nodes in increments of 10, calculating the average as the experimental result to compare the actual performance of the different algorithms. Throughput is calculated as follows:(8)$$TPS=\frac{N_{\Delta t}}{\Delta t},$$
where $N_{\Delta t}$ is the number of transactions completed within time Δt. The results are shown in Figure 9.

Analysis of the experimental images and data Table 7 reveals that as the number of nodes increases, the throughput of all four algorithms exhibits a downward trend. However, PBFT and GPBFT show a faster decline rate. Although CBFT demonstrates more stability than the other three algorithms, it suffers from slow startup times. Overall, AE-PBFT maintains a throughput that is approximately 24.7% higher than that of the closest competitor, GPBFT, indicating superior performance. This advantage is primarily due to the reduced delay associated with each transaction. Consequently, our solution can complete more transactions in a given timeframe, resulting in higher throughput compared to other solutions. Therefore, the approach proposed in this paper offers significant advantages in practical applications.

### 7.3. Communication Overhead

In this section, we will evaluate the communication overhead of the algorithm. Communication overhead refers to the total number of messages exchanged within the system when nodes execute the consensus algorithm. Generally, higher communication complexity correlates with lower consensus efficiency and increased resource consumption during operation. In the experiment, we fixed the number of requests at 200 and conducted 100 continuous tests with the number of participating nodes ranging from 0 to 100 in increments of 10, calculating the average as the result. The number of messages exchanged within the entire network for a single request is used to represent the communication overhead in the consensus process. The communication overhead is as follows:(9)$$N_{C}=\frac{N_{Message}}{n},$$
where $N_{Message}$ is the total number of messages exchanged in the entire network when completing *n* requests. The experimental results are shown in Figure 10.

The experimental results in Table 8 indicate that the communication overhead required by PBFT to achieve consensus is significantly higher than that of the other three algorithms. As the number of nodes increases—particularly when reaching 80 nodes—the communication overhead for PBFT escalates rapidly, while the overhead for the other algorithms remains relatively stable. From an average perspective, the trends for GPBFT and CBFT are similar; however, CBFT demonstrates superior performance. This is attributed to CBFT’s division of the consensus process into intra-group and inter-group consensus, which reduces the number of nodes involved in global consensus and minimizes the volume of information exchanged between nodes. AE-PBFT outperforms CBFT, achieving an average reduction in communication overhead of approximately 71.2%. This improvement is primarily due to the algorithm’s ability to swiftly update credit points, significantly decreasing the scale of nodes participating in consensus and thus reducing communication complexity while optimizing system performance in real-world environments.

### 7.4. Fault Tolerance

In this part of the experiment, we will evaluate the fault tolerance of the system. Fault tolerance refers to the ability of the system to withstand interference and achieve consensus in the presence of malicious nodes. A system with higher fault tolerance will reach consensus more quickly. To test this, we introduced a certain number of malicious vehicle nodes that send misleading messages to disrupt the consensus process. We fixed the total number of nodes participating in the consensus at 60 and gradually adjusted the proportion of malicious nodes from 0%, 5%, 10%, to 30%. We will measure the time required for nodes to reach consensus under varying degrees of interference, using this as a benchmark to assess their fault tolerance. A smaller consensus time indicates a higher level of fault tolerance. The fault tolerance is as follows:(10)$$FTA=\frac{1}{\bar{T_{F}}},$$
where $\bar{T_{F}}$ is the mean consensus delay of the algorithm under different proportions of malicious nodes. A lower consensus delay in the presence of interference indicates a stronger fault tolerance. The experimental results are shown in Figure 11 and Figure 12.

Analysis of the results reveals that as the proportion of malicious nodes increases, the consensus delay for PBFT rises sharply, while the consensus delays for the other three algorithms remain relatively stable. Compared to the best-performing CBFT, AE-PBFT demonstrates a delay that is an order of magnitude lower, indicating superior performance. Additionally, we define fault tolerance as the inverse of the average delay for each algorithm under interference, suggesting that a smaller value indicates stronger fault tolerance in the system. The results in Table 9 clearly show that AE-PBFT exhibits significantly greater fault tolerance than the other algorithms. This indicates that AE-PBFT can function effectively in an environment subjected to malicious interference. The rapid iteration of credit points enhances the efficiency of node screening, and the adaptive mechanism allows for quick responses and adjustments to external conditions, making it more resilient in harsh environments.

## 8. Potential Applications in Other Fields

### 8.1. Supply Chain Management

In supply chain management, the integration of blockchain and consensus algorithms, such as AE-PBFT can address key challenges, including transparency, traceability, and trust among stakeholders. The AE-PBFT algorithm, with its dynamic consensus group and trust acceleration mechanism, is particularly well-suited for supply chains that involve diverse participants, such as manufacturers, distributors, retailers, and consumers. By leveraging AE-PBFT, the system can dynamically adjust the consensus group ratio to adapt to varying levels of trustworthiness among participants, especially in complex multi-tiered supply chains. Malicious actors—such as those introducing counterfeit goods—can be quickly identified and excluded based on their historical behavior. Additionally, AE-PBFT’s efficient consensus mechanism ensures low latency and high throughput, making it feasible for real-time tracking of goods, reducing delays, and enhancing operational efficiency. This can be especially beneficial for industries like pharmaceuticals or food supply chains, where product authenticity and timely delivery are critical.

### 8.2. Healthcare

In healthcare, data security, privacy, and integrity are paramount, particularly when dealing with sensitive patient information and medical records. AE-PBFT can provide a robust foundation for blockchain-based healthcare systems by offering secure and efficient consensus mechanisms for managing electronic health records (EHRs), medical research data, and insurance claims. The node trust management model of AE-PBFT ensures that only trusted participants, such as hospitals, clinics, and insurance companies can access or contribute to the blockchain. Malicious nodes attempting to tamper with or misuse patient data are quickly identified and excluded, ensuring data integrity and patient privacy. Furthermore, the dynamic consensus group mechanism can adapt to varying network conditions, such as during large-scale medical emergencies, ensuring consistent performance and fault tolerance. AE-PBFT’s low communication overhead and high throughput make it ideal for real-time data sharing between healthcare providers, enabling seamless collaboration and improving patient care outcomes.

### 8.3. Digital Copyright Protection

In the realm of digital copyright protection, the proliferation of unauthorized copying and distribution poses significant challenges to content creators and rights holders. AE-PBFT can serve as a key component of a blockchain-based copyright management system by securely recording ownership, licensing agreements, and usage rights for digital assets such as music, videos, and software. The algorithm’s trust acceleration mechanism enables rapid differentiation between legitimate and malicious users, ensuring that only authorized entities can participate in the consensus process. For example, if a malicious node attempts to falsify ownership records or distribute unauthorized copies, its behavior will be quickly identified and mitigated. The dynamic consensus group mechanism can also optimize the system’s performance by adjusting to the network’s trust landscape, ensuring scalability and reliability in environments with a high number of participants, such as global content distribution networks. By leveraging AE-PBFT, digital copyright protection systems can provide immutable, transparent, and efficient rights management, helping to safeguard intellectual property and incentivize creativity.

## 9. Conclusions

In this paper, we propose a novel consensus algorithm, AE-PBFT, designed for application in the IoV system. The algorithm incorporates two core mechanisms. The first is the trust management model based on a gradual acceleration mechanism. This model adjusts the weight of credit changes for each node based on its continuous historical behavior, enabling the system to quickly distinguish malicious nodes from normal ones. By doing so, it significantly improves the efficiency of node screening and consensus completion. The second mechanism is the dynamic consensus group division model that adapts to environmental changes. This mechanism enhances the traditional consensus group approach by incorporating an environmental adaptive adjustment mechanism based on the overall credit mean of nodes. During operation, the consensus group ratio is dynamically adjusted according to the overall node credit distribution, enabling the system to better handle extreme scenarios and enhance reliability and fault tolerance. Experimental results demonstrate that AE-PBFT outperforms state-of-the-art consensus algorithms, including PBFT, GPBFT, and CBFT, in terms of consensus delay, throughput, communication overhead, and fault tolerance. Its advantages are particularly evident in extreme environments where malicious nodes constitute the majority. Furthermore, we propose a vehicle-service provider–RSU IoV system architecture based on the AE-PBFT algorithm. This architecture features a three-layer structure comprising a CA authentication layer, an edge interaction layer, and a data storage layer. The identity information of entities within the system is managed through registration and authentication, ensuring stable and secure system operation.

Future work will focus on optimizing the gradual acceleration mechanism in the existing credit model to achieve more refined tiered effects. Additionally, efforts will be directed toward improving the consensus group ratio algorithm to further enhance the environmental adaptability of the dynamic consensus group. Beyond IoV, the proposed framework has the potential for broader applications in fields, such as supply chain management, healthcare, and digital copyright protection, offering secure and efficient services across a wide range of industries.

## Figures and Tables

**Figure 1 sensors-25-01030-f001:**
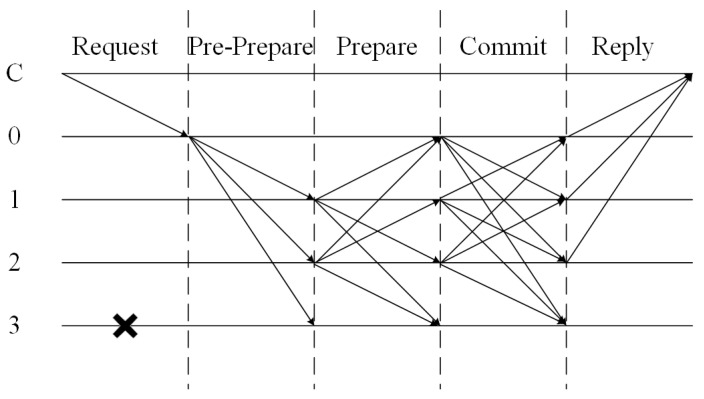
Consensus process of PBFT algorithm.

**Figure 2 sensors-25-01030-f002:**
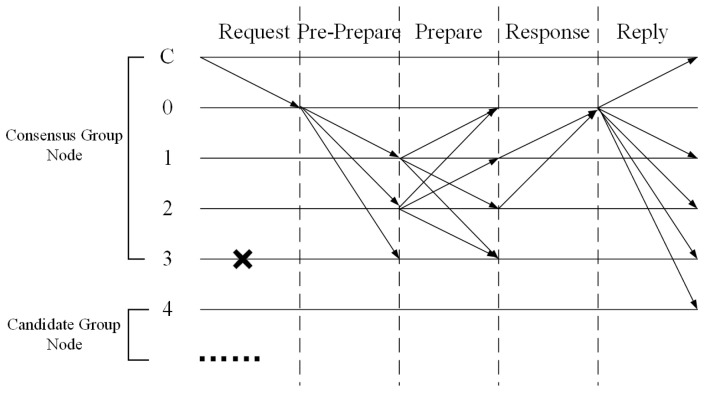
Consensus process of AE-PBFT algorithm.

**Figure 3 sensors-25-01030-f003:**
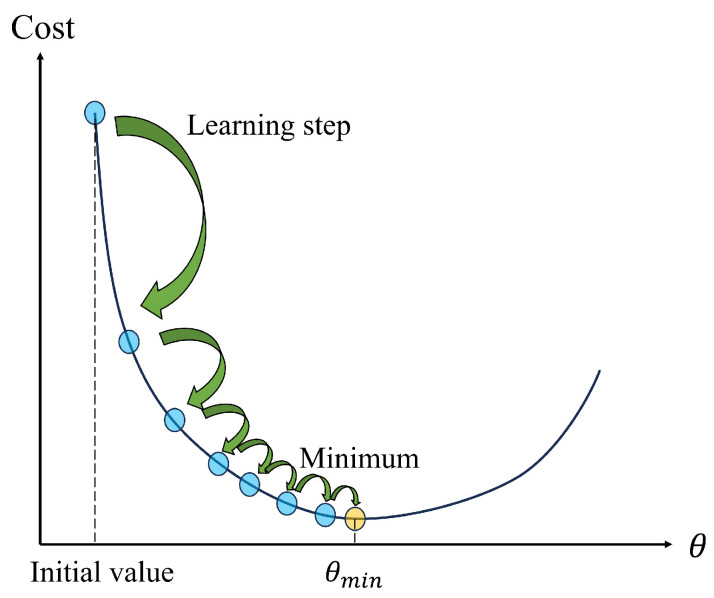
Model training speed under different learning rates.

**Figure 4 sensors-25-01030-f004:**
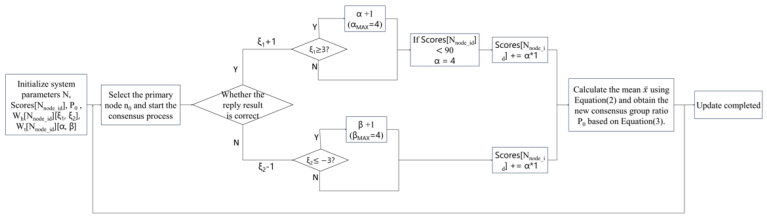
Trust management model based on gradual acceleration mechanism.

**Figure 5 sensors-25-01030-f005:**
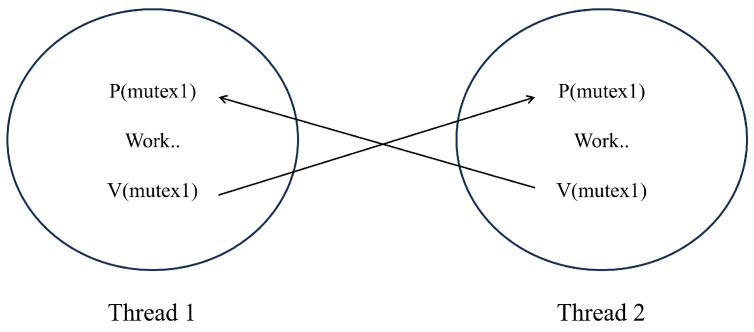
Recording semaphore mechanism.

**Figure 6 sensors-25-01030-f006:**
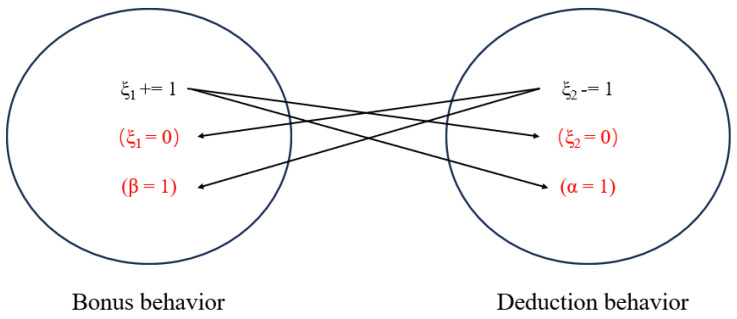
Simulated mutex semaphore mechanism.

**Figure 7 sensors-25-01030-f007:**
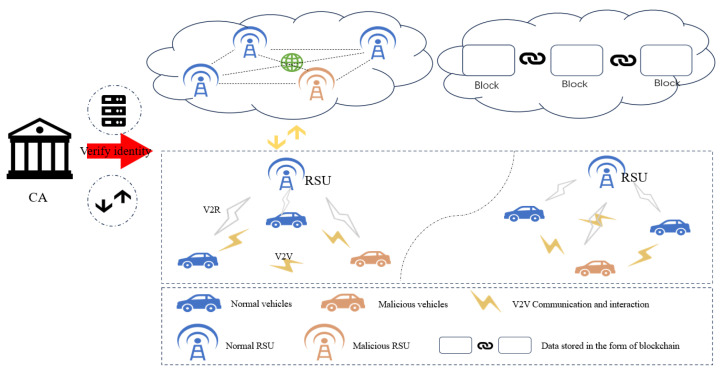
Construction of IoV system.

**Figure 8 sensors-25-01030-f008:**
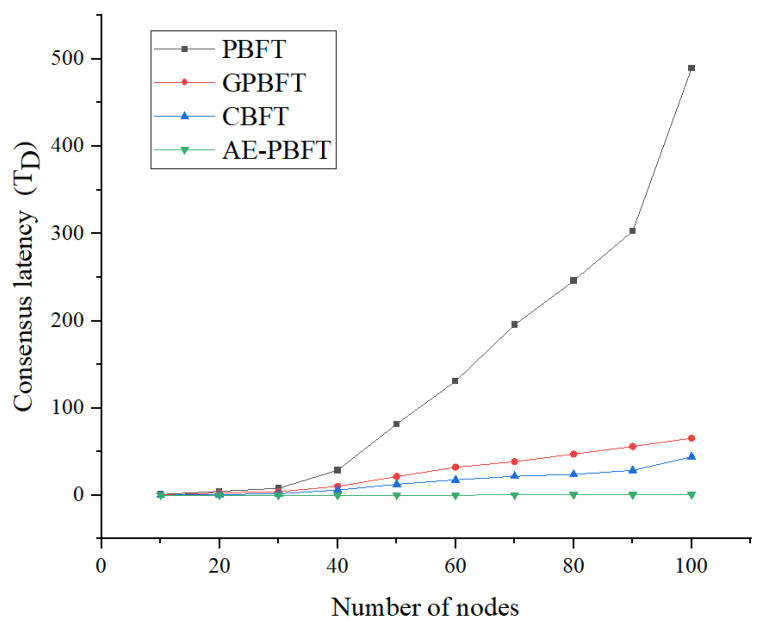
Consensus latency.

**Figure 9 sensors-25-01030-f009:**
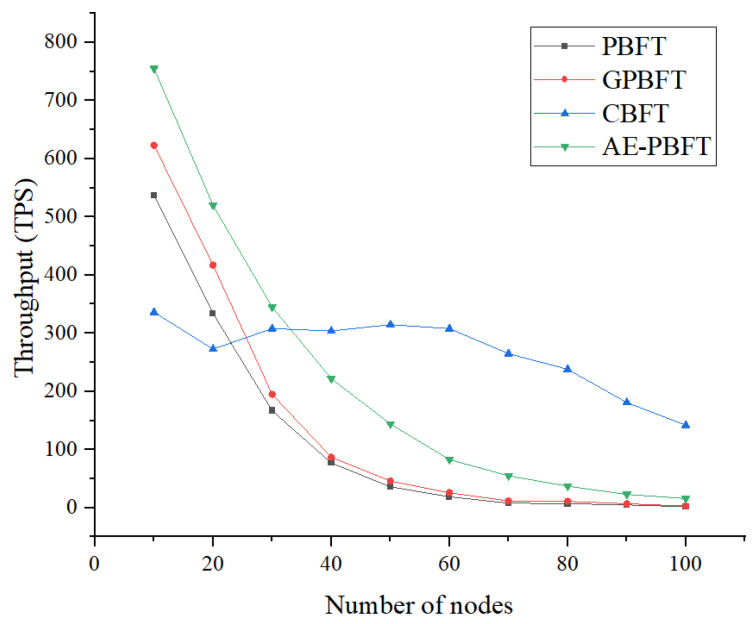
Throughput.

**Figure 10 sensors-25-01030-f010:**
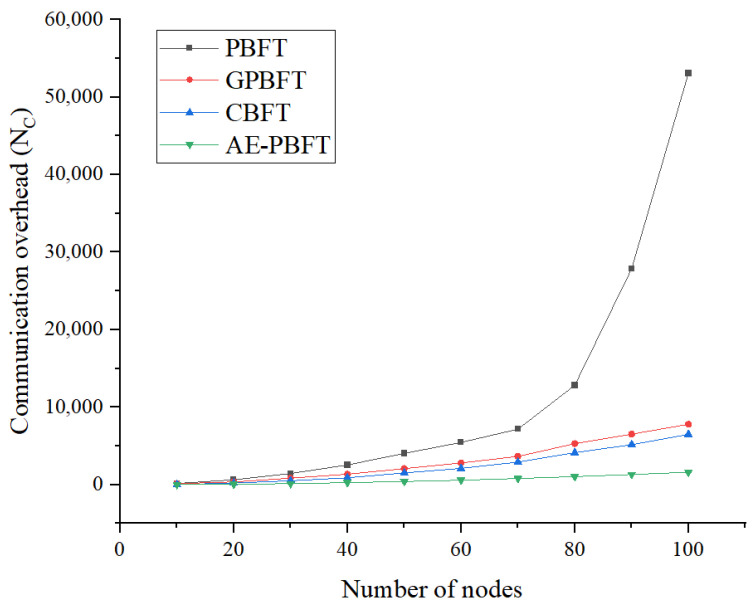
Communication overhead.

**Figure 11 sensors-25-01030-f011:**
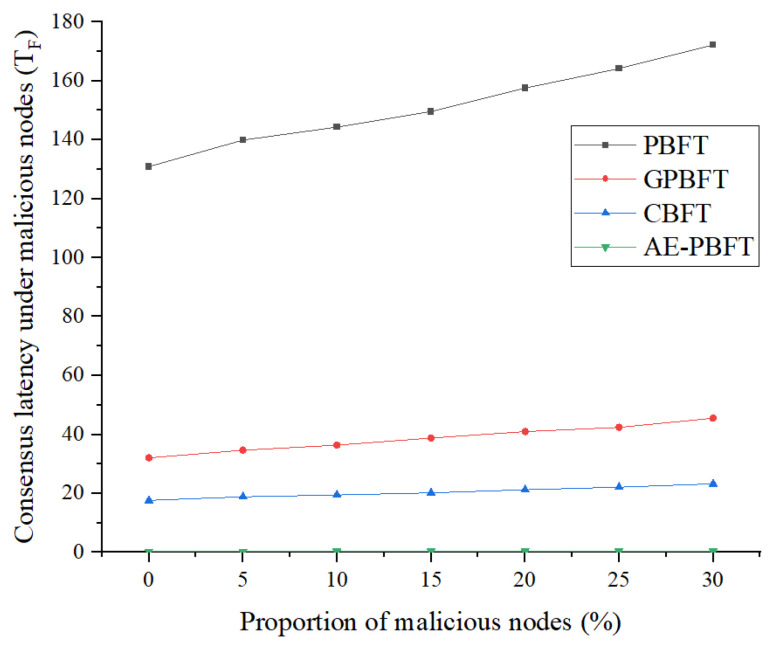
Consensus latency under malicious nodes.

**Figure 12 sensors-25-01030-f012:**
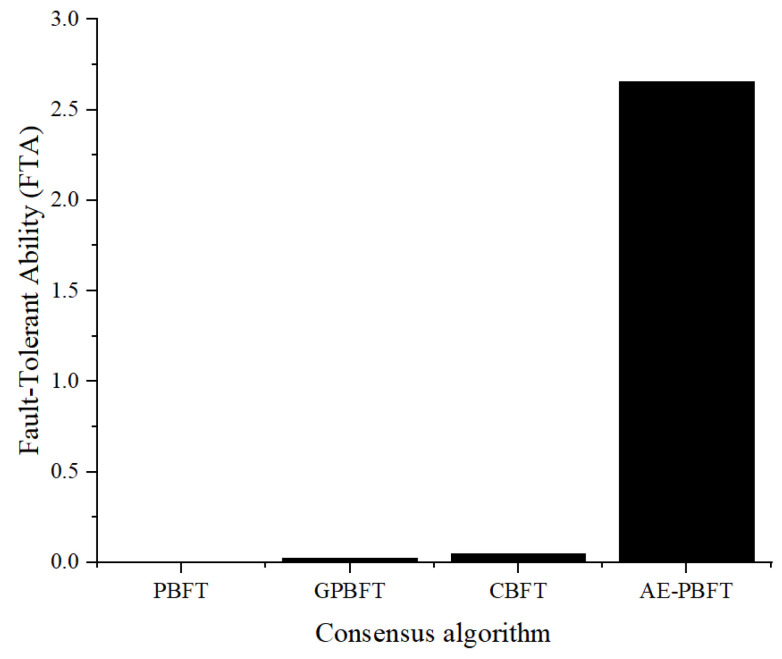
Fault-tolerance ability.

**Table 1 sensors-25-01030-t001:** Parameters used and the definitions.

Symbol	Definition
N	All system nodes
$N_{Node\_id}$	A system node with a corresponding id
$W_{h}$	Historical behavior counter
$\xi_{1}$,$\xi_{2}$	Parameters for recording continuous historical behavior
$W_{t}$	Trust coefficient recorder
α,β	Reward and penalty weight coefficient
Scores	Record the current credit score of all system nodes
$H_{Threshold}$	Low score compensation warning threshold
*P*	Consensus group proportion coefficient
$P_{0}$	Initial consensus group ratio coefficient
$N_{CO}$	Consensus Group Node
$N_{CA}$	Candidate group nodes
$a_{1}$,$a_{2}$	Weighting coefficient

**Table 2 sensors-25-01030-t002:** Gradient acceleration mechanism for continuous bonus behavior.

Consecutive Bonus Behaviors $\xi_{1}$	Reward Weight Coefficient α	Penalty Weight Coefficient β
[0,3)	1	No adjustments
[3,4)	2	No adjustments
[4,5)	3	No adjustments
[5,)	4	No adjustments

**Table 3 sensors-25-01030-t003:** Gradient acceleration mechanism for continuous score reduction behavior.

Consecutive Deduction Behaviors $\xi_{2}$	Reward Weight Coefficient α	Penalty Weight Coefficient β
(−3,0]	No adjustments	1
(−4,−3]	No adjustments	2
(−5,−4]	No adjustments	3
(,−5]	No adjustments	4

**Table 4 sensors-25-01030-t004:** Low score compensation mechanism.

Credit Score	Reward Weight Coefficient α	Penalty Weight Coefficient β
>=90	No adjustments	No adjustments
<90	4	No adjustments

**Table 5 sensors-25-01030-t005:** Communication complexity of four algorithms.

Consensus Algorithm	Communication Complexity
PBFT	$2n\left(n-1\right)$
GPBFT	$n^{2}+n-1$
CBFT	${\left(\frac{n}{m}\right)}^{2}-\frac{3n}{m}+2n$
AE-PBFT	$\approx \left(\frac{2n}{5}-1\right)\left(\frac{2n}{5}+1\right)$

**Table 6 sensors-25-01030-t006:** Consensus latency of different algorithms.

Number of Nodes	PBFT	GPBFT	CBFT	AE-PBFT
10	0.912718	0.218563	0.019014958	0.013005
20	4.692819	2.744698	0.983819	0.0393131
30	8.134397	4.1610796	1.7307227	0.086635
40	28.775688	10.2884976	6.0453126	0.13285766
50	81.806661	21.5135286	12.62448	0.210629
60	130.896491	32.104809	17.6887	0.31799066
70	195.557394	38.658515	21.898924	0.40999
80	245.676349667	47.2555075	24.003403	0.55243
90	302.76820014	56.00594428	28.6441059	0.729233
100	489.4821655	65.407161344	44.297028	0.92298

**Table 7 sensors-25-01030-t007:** Throughput of different algorithms.

Number of Nodes	PBFT	GPBFT	CBFT	AE-PBFT
10	537	623	336	755
20	334	417	273	520
30	167	195	308	345
40	77	87	304	222
50	36	46	315	144
60	19	26	308	83
70	8	12	265	55
80	7	11	238	37
90	5	7	181	23
100	2	3	142	16

**Table 8 sensors-25-01030-t008:** Communication overhead of different algorithms.

Number of Nodes	PBFT	GPBFT	CBFT	AE-PBFT
10	174	145	42	21
20	647	362	225	72
30	1449	853	513	155
40	2565	1373	908	271
50	4053	2090	1523	417
60	5458	2815	2116	595
70	7180	3663	2923	808
80	12,835	5291	4148	1043
90	27,883	6535	5178	1320
100	53,076	7801	6502	1622

**Table 9 sensors-25-01030-t009:** Consensus latency under malicious nodes.

Malicious Nodes Ratio	PBFT	GPBFT	CBFT	AE-PBFT
0	130.896491	32.104809	17.6887	0.3179906
5	139.926127	34.705288	18.9092203	0.325614
10	144.307101	36.438948	19.4929474	0.34115
15	149.60185	38.8147	20.21641532	0.377724
20	157.550822	40.9978296	21.2901193	0.396956
25	164.154401	42.44254	22.1816298	0.416435
30	172.214484	45.5567	23.2606405	0.460263

## Data Availability

Due to privacy issues, we are unable to provide data.

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
