# Peer review of "A Blockchain Solution for the Internet of Vehicles with Better Filtering and Adaptive Capabilities"

_sensors, 2025, doi:10.3390/s25041030_

Round 1
Reviewer 1 Report
Comments and Suggestions for Authors
The article is devoted to a relevant problem. Overall, the article is well written. However, there are a number of comments In Table 5, the formulas need to be corrected .
Tables 5, 7, 8 should be formatted.
Punctuation marks should be added to formulas.
Equation (1) needs comments. It is not specified whether the function is differentiable. Why are lowercase and uppercase letters used in this formula?
All abbreviations should be provided with expansions where they first appear.
• What is the main question addressed by the research?
The authors wanted to speed up the environment's adaptive consensus algorithm.
• Do you consider the topic original or relevant to the field? Does it
address a specific gap in the field? Please also explain why this is/ is not
the case.
• What does it add to the subject area compared with other published
material?
Section 2.3. (Advancements in Consensus Algorithms for Integrated Systems) discusses current developments on this topic
Lines 722 - 726: Compared to the best-performing CBFT, AE-PBFT demonstrates a delay that is an order of magnitude lower, indicating superior performance. Additionally, we define fault tolerance as the inverse of the average delay for each algorithm under interference, suggesting that a smaller value indicates stronger fault tolerance in the system. The results clearly show that AE-PBFT exhibits significantly greater fault tolerance than the other algorithms.
( comparison with other methods are shown in Table 8 and 9 and in Figs 11, 12, • Are the conclusions consistent with the evidence and arguments presented
and do they address the main question posed? Please also explain why this
is/is not the case.
Conclusions are made using experimental result (Table 8 and 9)
Сomparison with other methods are shown in Figs 11 and 12, Tables 8, 9
• Are the references appropriate?
Paper [29] Memon, I.; Shaikh, R.A.; Shaikh, H. Dynamic pseudonyms trust-based model to protect attack scenario for internet of vehicle 874 ad-hoc networks. Multimedia Tools and Applications 2024, 83, 13395–13426 is RETRACTED ARTICLE (Retraction Note: Multimedia Tools and Applications (2024) 83:13395-13426 https://doi.org/10.1007/s11042-023-16110-5.The Editor-in-Chief and the publisher have retracted this article. An investigation by the publisher found a number of concerns, including but not limited to citations which do not support claims made in the text, non-standard phrasing, and image irregularities. Based on the investigation’s findings the Editor-in-Chief therefore no longer has confidence in the results and conclusions of this article.
Author Imran Memon has stated that the authors disagree with this retraction.
https://www.researchgate.net/publication/382967414_Retraction_Note_Dynamic_pseudonyms_trust-based_model_to_protect_attack_scenario_for_internet_of_vehicle_ad-hoc_networks • Any additional comments on the tables and figures. Tables 5,7,8 need to be formatted and some formulas need to be corrected. The formulas in Table 5 are not written in mathematical style. The text in Figure 4 should be larger.
Reviewer 2 Report
Comments and Suggestions for Authors
The paper presents a blockchain-based solution for the Internet of Vehicles (IoV) with a focus on improved filtering and adaptive capabilities. While the topic is highly relevant and the proposed approach shows potential, the manuscript lacks a thorough analysis of the limitations of existing blockchain technologies in the context of IoV. Specifically, the following critical issues are not adequately addressed:
1. Add a dedicated subsection in the Introduction discussing the limitations of existing blockchain technologies in IoV, focusing on performance, scalability, and adaptability.
2. In Sections 1 and 2, the authors frequently use the abbreviation PBFT without providing its full form or a brief explanation of its meaning. Add a sentence or footnote when PBFT is first introduced, explaining its full form and its significance in the context of the proposed solution.
3. In Section 4, the discussion on the learning rate appears to be overly basic and does not contribute significantly to the overall value of the manuscript. The concept of the learning rate is a fundamental aspect of machine learning and optimization algorithms, and it is likely to be well-understood by the target audience of this journal.
4. In the comparative analysis in Section 5, the authors mention two algorithms, GPBFT and CBFT, as baseline methods for comparison. However, the manuscript does not provide any background information about these algorithms. Why were GPBFT and CBFT specifically selected as baselines? Are they state-of-the-art methods or representative of a particular class of solutions? The authors should clarify the rationale behind their choice of comparison algorithms.
5. The manuscript contains some formatting issues related to the presentation of tables and figures that need to be addressed to improve readability and consistency:
Tables 1, 4-9 Font Size: The font size in Tables 1, 4-9 appears to be too large, making it visually inconsistent with the rest of the manuscript. The font size of tables should match the body text size to maintain a professional and uniform appearance.
Figure 4 Size: Figure 4 is currently too small, which may make it difficult for readers to interpret the details. The figure should be resized to ensure that all elements (e.g., labels, data points, annotations) are clearly visible.
Comments on the Quality of English LanguageThe English could be improved to more clearly express the research.
Round 2
Reviewer 2 Report
Comments and Suggestions for Authors
The authors have made significant improvements to the manuscript based on the previous review comments. All concerns almost have been addressed, and the revised manuscript is now more comprehensive, clear, and well-organized.
Author Response
Dear Reviewer,
Thank you for your valuable comments, which are very helpful to improve the quality of the paper. I wish you a Happy New Year and all the best in your future work!
Kind regards,
Ma Runyu